# Increased Presence of Complement Factors and Mast Cells in Alveolar Bone and Tooth Resorption

**DOI:** 10.3390/ijms22052759

**Published:** 2021-03-09

**Authors:** Kathrin Luntzer, Ina Lackner, Birte Weber, Yvonne Mödinger, Anita Ignatius, Florian Gebhard, Susann-Yvonne Mihaljevic, Melanie Haffner-Luntzer, Miriam Kalbitz

**Affiliations:** 1Center for Trauma Research Ulm (ZTF), University of Ulm, 89081 Ulm, Germany; kathrin@kathrin-widmann.de (K.L.); ina.lackner@uni-ulm.de (I.L.); birte.weber@uni-ulm.de (B.W.); yvonne.moedinger@gmx.de (Y.M.); Anita.Ignatius@uni-ulm.de (A.I.); florian.gebhard@uniklinik-ulm.de (F.G.); Melanie.haffner-luntzer@uni-ulm.de (M.H.-L.); 2Department of Traumatology, Hand-, Plastic-, and Reconstructive Surgery, University Medical Center, 89081 Ulm, Germany; 3Small Animal Clinic Ravensburg Evidensia GmbH, Eywiesenstraße 4, 88212 Ravensburg, Germany; 4Institute of Orthopedic Research and Biomechanics, University of Ulm, 89081 Ulm, Germany; 5Im Andermannsberg 45, 88212 Ravensburg, Germany; susann@mihaljevic-rv.de

**Keywords:** tooth resorption, periodontitis, mast cells, complement system, osteoclasts

## Abstract

Periodontitis is the inflammatory destruction of the tooth-surrounding and -supporting tissue, resulting at worst in tooth loss. Another locally aggressive disease of the oral cavity is tooth resorption (TR). This is associated with the destruction of the dental mineralized tissue. However, the underlying pathomechanisms remain unknown. The complement system, as well as mast cells (MCs), are known to be involved in osteoclastogenesis and bone loss. The complement factors C3 and C5 were previously identified as key players in periodontal disease. Therefore, we hypothesize that complement factors and MCs might play a role in alveolar bone and tooth resorption. To investigate this, we used the cat as a model because of the naturally occurring high prevalence of both these disorders in this species. Teeth, gingiva samples and serum were collected from domestic cats, which had an appointment for dental treatment under anesthesia, as well as from healthy cats. Histological analyses, immunohistochemical staining and the CH-50 and AH-50 assays revealed increased numbers of osteoclasts and MCs, as well as complement activity in cats with TR. Calcifications score in the gingiva was highest in animals that suffer from TR. This indicates that MCs and the complement system are involved in the destruction of the mineralized tissue in this condition.

## 1. Introduction

Oral diseases, including periodontitis, are a severe global clinical problem. Periodontitis is the inflammatory destruction of the tooth-surrounding and -supporting tissue, eventually resulting in tooth loss. It affects between 3 and 50% of the human population depending on the definition criteria [1]. Gingivitis is an inflammatory process, which affects the gingiva alone. Its characteristics are a normal alveolar margin and architecture, without attachment loss but inflamed gums [2]. In most cases, the gingival inflammation is induced by the immune response to oral bacteria located along the neck of the tooth. However, this inflammation can progress to periodontitis. Here, the inflammatory response affects the tooth-surrounding tissue, leading to periodontal pockets [3]. Consequently, oral bacteria accumulate in these pockets, where the host’s immune response is reinforced and the destruction of the surrounding tissue is amplified.

Another aggressive disease of the oral cavity is tooth resorption (TR), a dental condition associated with the destruction of the dental mineralized tissue (enamel, dentine, cementum). TR is observed in several species, including cats, dogs and humans [4,5]. The prevalence of TR in humans varies considerably, depending on the considered criteria and population: Tsesis et al. evaluated radiographically the prevalence of TR in different tooth groups in a Middle Eastern population, and found teeth exhibiting TR in 28.8% of subjects [6]. Haapasalo et al. suggest a prevalence of between 0.01% and 1% (patients affected) for internal inflammatory root resorption, which is a relatively rare kind of resorption [7]. Motokawa et al. found root resorption, induced by orthodontic treatment, in 78.2% of the orthodontic patients [8]. The occurrence in cats is remarkably high: TR affects 26–72.5% of all cats and the prevalence further increases with age [9,10,11,12]. Therefore, cats were used in the present study as a naturally occurring disease model. In both, cats and humans, TR etiology remains unclear. Recently, as in periodontitis, the immune system has been considered as being an important factor in the development of resorptive lesions [13,14,15].

The present study was performed to further investigate the presence of cells and immune components, which might play an important role in alveolar bone and dental mineralized tissue destruction. Here, we focused on osteoclasts, mast cells (MCs) and the complement system. Furthermore, the presence of calcium crystals in the TR pathophysiology was evaluated. Osteoclasts are the cells responsible for the removal of mineralized tissue. Their formation and activity is regulated by several mechanisms. Ignatius et al. identified the complement factors C3a and C5a, in synergism with interleukin (IL)-1β, as modulators in osteoclast formation [16]. Kroner et al. recently showed that the stimulation of MCs with complement C5a results in a degranulation of pre-performed granule-embedded mediators as well as the release of the novo-synthesized cytokines [17]. These released mediators led to an increased number of osteoclast-like cells. Moreover, the inflammatory response in crystallopathies is triggered by the activation of the complement system, as was previously shown in gout research [18]. A crystallopathy termed primary hyperoxaluria is characterized by the deposition of calcium crystals in several tissues and is known to be associated with tooth resorption, because of the inflammatory process linked to gingival crystal deposition [19]. These preliminary findings have led us to the question, whether MCs and the complement factors C3 and C5, as well as calcium crystals are critically involved in alveolar bone and dental mineralized tissue destruction. To answer this question as precisely as possible, we not only compared the occurrence of MCs, complement factors and calcium crystals in controls, periodontitis and TR, but also in gingivitis, which can progress to periodontitis. 

## 2. Results

### 2.1. Analysis of Anamnestic Data 

Blood, teeth and gingiva samples were collected from cats either affected by gingivitis, periodontitis or TR, during an appointment for routine dental treatment under anesthesia. Breeds and non-breeds aged between 10 months and 18 years were included. Each cat had been neutered. An owner survey was completed for each cat. In 4 of 24 cases, the patients’ owners did not notice any symptoms at home. They came for dental treatment on the veterinarian’s advice. The most common symptom, which was noticed by the other owners, was foetor ex ore. Apart from those cats, which only suffered from gingivitis, the owners also noticed in some cases pain during feed intake, avoiding of dry food, inappetence and/or weight loss.

Statistical analysis revealed that there was no significant difference between the three groups in age, indoor only or the feeding but a significant difference between the gingivitis and periodontitis groups regarding the gender (*p* ≤ 0.05) (Table 1).

Additionally, the same samples were collected from four healthy controls (two neutered males and two neutered females; between 1 and 2 years old). These controls were born as specific pathogen free (SPF) cats, though they grew up with a conventional keeping at Bayer AG, Monheim-on the-Rhine, North Rhine-Westphalia, Germany.

### 2.2. Calcium Crystals

As calcium crystals were shown to contribute to the TR pathomechanisms in humans suffering from primary hyperoxaluria [19], we identified them in histological sections of the feline gingiva. In the control and gingivitis groups, most samples had no or only rare tiny crystals. By contrast, in the TR group, most samples exhibited either many crystals or very large ones (Figure 1A). There was a significant difference between the TR group and both the control (*p* ≤ 0.05) and gingivitis groups (*p* ≤ 0.05) and a tendency for a difference between the gingivitis and periodontitis groups (Figure 1B).

### 2.3. Osteoclasts

Tartrate-resistant acid phosphatase (TRAP) staining of the histological tissue sections was applied to identify osteoclasts in the bone and at the surface of teeth affected by either periodontitis or TR. Only these two groups were evaluated, because the other groups did not display signs of alveolar bone or tooth loss in the clinical radiographic examination. TRAP-positive multinucleated cells were present on both, the bone and tooth surfaces (Figure 2B). Osteoclasts could be identified in both, the periodontitis and TR groups, with the number of TRAP-positive cells at the tooth and bone surfaces being significantly higher in the TR group (osteoclasts/mm tooth surface: *p* ≤ 0.05; osteoclasts/mm bone surface: *p* ≤ 0.01). There was also a significantly greater percentage of eroded tooth surface in the TR group (*p* ≤ 0.01) (Figure 2C).

### 2.4. MCs

MCs were detected and quantified by immunohistochemical staining for histamine (Figure 3A). The number of MCs in the TR group was significantly higher compared to the gingivitis group (*p* ≤ 0.01). Furthermore, the staining revealed a significant increase of MCs in the TR group compared with the control group (*p* ≤ 0.01). There was no difference between the TR and periodontitis groups. Moreover, there was also no difference between the control and gingivitis groups (Figure 3B). 

### 2.5. Complement Receptor C5a

The expression of C5a receptor (C5aR) in the epithelial and endothelial layers, as well as in the lamina propria was evaluated by an immunohistochemical staining (Figure 4A) and each scored 0 to 3 (0 = very rare expression, 1 = mild expression, 2 = moderate expression, 3 = high expression). No significant difference between the four groups could be determined. Nevertheless, there was a small, but non-significant increase in the C5a receptor expression in the endothelial layer in the periodontitis and TR groups compared with the control and gingivitis groups (Figure 4B). 

### 2.6. Complement Factor C5a

C5a-positive cells were identified by IHC in gingiva samples (Figure 5A). From the C5a cell count there was a trend for increased C5a-positive cells in the TR group compared with the gingivitis group (*p* = 0.1168) (Figure 5B). Furthermore, double immunofluorescence staining was performed to determine whether MCs in the gingiva were positive for C5a. This staining revealed MCs positive for C5a (Figure 5C). Due to the incompatibility of the anti-histamine antibody and the anti-C5a antibody, Avidin-Texas Red was used as a specific mast cell granule staining. Specificity of Avidin staining was proven by staining of bone marrow sections from mast cell competent wildtype mice and Mcpt5-Cre^+^ R-DTA^flox/flox^ mast cell deficient mice (Appendix A).

### 2.7. Complement Factor C3

C3-positive cells were identified by IHC in gingiva samples (Figure 6A). The C3-positive cell count was significantly increased in the TR group compared with the control, gingivitis and periodontitis groups (*p* ≤ 0.05). There was no difference between the control, gingivitis and periodontitis groups (Figure 6B). Furthermore, a double immunofluorescence staining was performed to determine whether MCs in the gingiva were positive for C3. This staining revealed MCs positive for C3 (Figure 6C). 

### 2.8. CH-50 and AH-50 Assays 

The CH-50 is a screening assay for the activation of complement classical pathway function, whereas the AH-50 is a screening assay for the activation of complement alternative pathway function. The assays revealed a significant difference in both the classical (Figure 7) and alternative pathways (Figure 8): The lysis capacity of the TR group was significantly higher than in the control group, which indicated that these cats exhibited significantly higher complement factor levels than the controls.

## 3. Discussion

This study was performed to reveal the involvement of MCs, the complement system and osteoclasts in periodontitis and TR. Furthermore, the presence of calcium crystals in the pathophysiology of TR was evaluated. 

Osteoclasts are large, multinucleated cells, which are responsible for bone resorption in physiological and pathological conditions. Moreover, it is well established that these cells are finally responsible for the tooth resorption process [20]. This point could be confirmed by the present study: As expected, osteoclasts were identified to be the cells associated with the resorption process in periodontitis and TR. Furthermore, the number of osteoclasts on the tooth and bone surfaces was significantly higher in the TR group than in the periodontitis group. The formation and activity of osteoclasts are regulated by several mechanisms. To further investigate possible triggers of increased osteoclast formation in periodontitis and TR, we examined the presence of MCs and of the complement factors C3 and C5a, as well as the C5aR expression and the deposition of calcium crystals in the gingiva. Von Kossa staining revealed that calcium crystals were present in the gingiva sections, particularly in those teeth affected by TR. It is known that humans who suffer from primary hyperoxaluria, a disease characterized by the deposition of calcium crystals in several tissues, display root resorption. This resorption process is associated with the gingival crystal deposition [19]. The inflammatory process in crystallopathies, however, is triggered by the activation of the complement system, as was shown previously in gout research [18]. The complement anaphylatoxins are strong chemokines for, among others, MCs [16,21]. Therefore, the deposition of calcium crystals in the gingiva might induce an immune reaction, finally leading to bone and tooth resorption. However, calcium deposition could also be the result of inflammatory processes in the gingiva. It is known that chronic inflammation is frequently associated with dystrophic calcium deposition, which is caused by a passive precipitation because of cell degeneration and necrosis [22]. 

MCs are known as key effector cells and modulators in the innate and acquired immune responses [23]. Furthermore, they can promote osteoclast formation and activity [17] and were shown to be increased in feline periodontal disease and TR [13]. Additionally, in the present study, the MC count was increased in these diseases. However, there was no difference in the occurrence of MCs in TR compared to periodontitis. Therefore, this cell type might play a role in both, alveolar bone and dental mineralized tissue destruction. 

Some human MCs express receptors for the complement factors C3a and C5a. An activation of these cells by the binding of these anaphylatoxins at their surface receptors results in degranulation and the production of cytokines and chemokines [24]. Such complement receptor-expressing MCs produce both chymase and tryptase [25]. Furthermore, human mast cell β-tryptase is able to cleave C3 and C5. In this way, the MC and the complement pathway are coupled. Furthermore, the working group of Fukuoka showed, that human skin MCs express the complement factors C3 and C5 [26]. In our study, double immunofluorescence staining was conducted to determine whether MCs in the feline gingiva express C3 and C5a. Cells could be identified that fluoresced both red and green, indicating, that feline MCs in the gingiva express C3 and C5a. Furthermore, several studies over recent decades demonstrated that the complement’s functions go beyond its role as a sensor and effector system. Local and frequently non-canonical complement activation influences all types of immune and non-immune cells [27]. For example, C3a and C5a are known to play a role in inflammatory bone disorders [28], including periodontitis [16,29,30,31,32]. Both anaphylatoxins in synergism with IL-1β induce a pro-inflammatory response in osteoblasts, as well as the production of the receptor activator of NF-κB Ligand, which induces osteoclast formation [32]. The inhibition of either C5aR or C3 in murine periodontitis models led to decreased inflammatory levels and tissue destruction [31,33]. In the present study, we showed a significantly increased number of C3-positive cells in the TR group compared to the control, gingivitis and periodontitis groups. No significant difference was found in the number of C5a-positive cells, because of the wide deviation within the periodontitis group. However, there was a trend for increased positively stained cells in periodontitis and TR compared with the other two groups. The wide deviation within the periodontitis group might be the result of the fact, that this group included both mild (mild horizontal bone loss) and severe forms of periodontitis (severe attachment loss between tooth and alveolar bone). However, a differentiation of the periodontitis group into subgroups was not possible because of the small group size. Moreover, the distinct dental biofilm of the cats included in the study might have contributed to the heterogeneity of this group. The dental biofilm as well as the immune reaction of the host are the main etiological factors for periodontal diseases. Under physiological conditions, there is a symbiotic relationship between the host’s immune reaction and the resident microorganisms [34]. Both, the colonization of the oral cavity with specific “periodontitis-associated bacteria” and an inadequate immune response of the host to the oral biofilm, mark the initial steps of the development of periodontal disease. 

The receptor for the complement anaphylatoxin C5a (C5aR) is not only expressed by some MCs, but also by several other cell types, including epithelial and endothelial cells [35]. In the present study, we investigated the C5aR expression of the epithelial layer, of the endothelial cells and of cells of the lamina propria. No significant difference between the four groups could be determined regarding the C5aR expression in each of these compartments. With regard to the epithelial layer, this result was expected, considering the receptor’s expression as part of the physiological, local immune response to the oral microbiome. By contrast, a noticeable difference in receptor expression regarding the endothelial layer was observed: increased C5aR expression was found in periodontitis and TR compared with the other two groups, indicating that this receptor might play a critical role during the inflammatory process. Furthermore, there was a difference in the positively stained cells of the lamina propria between the control group and the other groups. 

In accordance with this, the CH-50 and AH-50 assays revealed, that the systemic lysis capacity of the TR group was significantly higher than of the control group, which indicated that the cats affected by TR exhibited significantly higher systemic levels of complement components. This might be directly related to the dental disorder. It is already known that a specific single nucleotide polymorphism of complement factor C5 (rs17611) is associated with increased C5 serum levels [36]. This polymorphism was shown to be more prevalent in the group of periodontitis patients than in the healthy controls. However, because our controls were maintained under specific experimental conditions, we cannot rule out that there was a lack of stimulation of immune cells as well as other complement-producing cells, which led to a lower complement factor production in these animals. When this was the case, it could be assumed that TR is more likely to be a local inflammatory process driven by the production of complement factors by local immune cells, rather than a systemic dysfunction of the complement system. 

One limitation of our study is that for the immunohistochemical staining there were only males in the gingivitis group, whereas there were mainly females in the periodontitis group. Therefore, there was a significant difference regarding the gender distribution of these groups. Consequently, we cannot rule out a possible influence of sex hormones on our results. However, because each cat included in this study was neutered, we assume that sex hormones were not responsible for the detected differences between the groups.

Another limitation is that no mechanistic data are available. This is only an observational study. However, other studies strongly suggested an involvement of MCs in osteoclastogenesis [16,17]. In addition, the role of the complement system in bone homeostasis and disorders is well explored [37]. However, nothing is known about its possible involvement in TR. The results of the present study strongly suggest that the complement system plays a role in this dental disease. Further research is needed to confirm this hypothesis and rule out possible pathologic mechanisms leading to higher complement levels, locally and systemically. 

The cat, as an animal model, was used intentionally because the prevalence of TR in these animals is noticeably high (TR affects 26–72.5% of all cats and the prevalence further increases with age [9,10,11,12]). In research, it is frequently criticized that highly standardized mice are used and, therefore, the results might not be biologically relevant in more heterogeneous populations like humans. The cats used in the present study represent a highly diverse population therefore we believe that obtaining significant results from these cohorts might be more reflective of the actual biological situation. 

The present study is intended to be the starting point for new therapeutic options for TR not only in cats, but also in humans. The results suggest that the application of histamine or complement blockers in feline TR may prove useful. In general, the usage of complement blockers in inflammatory diseases is highly promising, but also potentially harmful. In particular, in the mouth, a functioning immune system is the prerequisite for a healthy balance between the host and the naturally occurring oral bacteria. Therefore, further research is needed on this topic.

## 4. Materials and Methods 

### 4.1. Sample Collection and Clinical Grading

The teeth, gingiva samples and blood were collected from 33 cats, which were between 10 months and 18 years old. In this clinical study we only used material which, from a veterinary point of view, had to be removed as part of the treatment or which could optionally be removed. Teeth were removed where necessary and gingiva adjacent to the affected tooth was collected. As part of anesthesia management, a vein catheter was inserted, so we could withdraw blood from some animals (Table 2). Breeds (Maine Coons, a Persian, a Ragdoll, a British Shorthair, a Siamese and a Birman) and non-breeds were included. The cats had an appointment for routine dental treatment under anesthesia at an animal hospital Evidensia GmbH in Ravensburg, Germany, between June 2017 and September 2018. The owners of the cats were asked whether they agreed that clinical parameters and remaining tissue could be used for research purposes. The owners’ written informed consent was obtained. According to the regulations of the Independent Local Ethics Committee of the University of Ulm, there was no human involvement in the present study, because no personalized data of health or sexuality of humans were recorded. The oral health status of the cats was determined by probing for periodontal pockets and by a full-mouth radiographic exam. Based on these clinical findings, we categorized each tooth and gingiva sample according to gingivitis, periodontitis or TR. Based on the anamnestic data, cats were assigned to the group with the most severe disease (gingivitis-periodontitis-TR) when showing more than one of these dental disorders.

Furthermore, the same samples were collected from four healthy controls (two neutered males and two neutered females; between 1 and 2 years old). These controls were born as SPF cats, though they grew up under conventional keeping at Bayer AG, Monheim on the Rhine, North Rhine-Westphalia, Germany. The study procedures adhered to the current national legislation (German protection of animals act (18 May 2006), last amended by article 4 paragraph 87 (BGBl. I S. 1666) on 18 July 2016 and the EU directives 63/2010 (on the protection of animals used for scientific purposes). All animal experiments were approved by the local and governmental review boards (State Agency for Nature, Environment and Consumer Protection (LANUV) of the State of North Rhine-Westphalia). 

### 4.2. Micro-Computed Tomography (µCT) Analysis

The teeth were analyzed using a µCT scanning device (Skyscan 1172, Billerica, MA, USA) operating at a resolution of 8 µm and a voltage of 50 kV and 200 mAV to determine tooth integrity and mineralization. µCT analysis was performed with the three-dimensional analysis software from Skyscan (NRecon, CTVox). Tooth mineral density was thresholded with a custom-made transfer function in CTVox. Highly mineralized zones were illustrated in yellow, while lower mineralized zones were illustrated in red. When signs of TR were visible in the µCT scans, teeth were assigned to the TR group independently from the clinical classification.

### 4.3. Histology and IHC

The gingiva samples were fixed immediately in 4% buffered neutral formalin for 48 h before being embedded in paraffin wax. The teeth were similarly fixed in formalin and subsequently decalcified in Ethylenediaminetetraacetic acid (EDTA) for 3 months before being embedded in paraffin. Sections of 8 µm were processed for histological and immunohistochemical analyses. TRAP staining was performed to identify osteoclasts as described previously [38]. Cells that were TRAP positive, multinucleated and associated with either the bone or the tooth surface were defined as osteoclasts. The relative number of osteoclasts was calculated per mm individually for the bone and tooth surface.

In the gingiva samples, calcium crystals were detected by von Kossa staining in paraffin-embedded gingiva samples. The solutions and reagents used were 1% aqueous silver nitrate solution, 5% sodium thiosulfate and 0.1% nuclear fast red solution. The sections were deparaffinized and hydrated to water and rinsed in several changes of distilled water. Subsequently, the sections were incubated with 1% silver nitrate solution under ultraviolet light for 20 min. Following a washing step with distilled water, un-reacted silver was removed with 5% sodium thiosulfate. Another washing step was performed, followed by counterstaining with nuclear fast red for 5 min and a further washing step. Dehydration was performed through graded alcohol and xylene. The sections were analyzed by light microscopy. The occurrence of crystals was evaluated using a scoring system of 0 to 3 (0 = no crystals; 1 = occasionally, tiny crystals; 2 = a few crystals; 3 = many crystals or very large ones).

Histamine (1:250, rabbit polyclonal antibody to histamine; Abcam ab37088, Cambridge, UK), complement factor C3 (1:75, mouse monoclonal antibody to activated C3, Hycult biotech HM2168, Uden, Netherlands), complement factor C5a (1:50, rabbit polyclonal antibody to C5a; Abbiotech 250565, Escondido, CA, USA) and C5a receptor (1:150, rabbit polyclonal antibody to C5R1; Abcam 59390) were detected by immunohistochemical staining. Sections were deparaffinized, rehydrated and blocked with 10% goat serum for 30 min at room temperature (RT) followed by incubation with the primary antibody for 2 h at 37 °C. Following a washing step with distilled water, the sections were incubated with the secondary antibody for 30 min at RT (C3: goat anti-mouse biotinylated (1:200, Invitrogen, Darmstadt, Germany), histamine, C5a and C5aR: goat anti-rabbit biotinylated (1:200 for C5a and C5aR/1:300 for histamine, Invitrogen)). Vectastain Elite ABC kit and Vector NovaRED substrate (both Vector Laboratories, Burlingame, CA, USA) were applied as per manufacturer’s specification for signal detection. The sections were counterstained using hematoxylin (Waldeck, Muenster, Germany). Species-specific isotype controls were used to confirm antigen-specific staining. The cell counts were performed by light microscopy. In sections labelled for histamine, C3 and C5a, the total number of positively labelled cells in each epithelial layer of the gingiva was counted. The relative number was calculated per mm^2^ of gingival area.

The evidence for C5a receptor-expressing cells was evaluated using a scoring system. Three compartments were considered: C5a-receptor expression of the epithelial layer, of the endothelial cells and the expressing cells in the lamina propria. These compartments were scored 0 to 3 (0 = very rare expression, 1 = mild expression, 2 = moderate expression, 3 = high expression).

Double immunofluorescence staining was conducted to determine whether MCs in the gingiva expressed C3 and C5a. Mast cell granules were stained using Avidin conjugated to Texas Red (1:200 ThermoFisher, Waltham, MA, USA) according to established protocols (Bergstresser et al., 1984). C3 and C5a were detected with the primary antibodies mentioned above and secondary antibodies goat-anti mouse AlexaFluor 488 (1:100 Abcam) and donkey anti-rabbit AlexaFluor 488 (1:100 Life Technologies, Carlsbad, CA, USA). Counterstaining was performed by Hoechst nucleus staining. Specificity of Avidin staining was proven by by staining of bone marrow sections from mast cell competent wildtype mice and Mcpt5-Cre^+^ R-DTA^flox/flox^ mast cell deficient mice (Appendix A). Femurs from these mice were fixed in 4% formalin for 48 h, decalcified and embedded in Paraffin. Then, 8 µm longitudinal sections were used for establishment of the Avidin staining. Sections from mast cell deficient mice were used as a biological negative control to analyze specificity of Avidin staining for mast cell granules (animal license approved by Regierungspäsidium Tübingen, O.135-7). 

### 4.4. Histomorphometric Analysis

The gingiva, the alveolar bone and the gingiva directly associated with the extracted teeth, as well as the teeth were evaluated by histomorphometric analysis. Image-analyzing software was used to measure the gingiva area (MMAF 1.4.0, Leica, Wetzlar, Germany). Measurement of the tooth and bone surface lengths, as well as the length of the osteoclast-eroded surface was performed using osteomeasure software (4.1.0.0, OsteoMetrics, Decatur, GA, USA). 

### 4.5. CH-50 Assay

For the CH-50 assay, sheep red blood cells (SRBC) pre-coated with rabbit anti-sheep red blood cell antibody (Complement Technology, Inc., Tyler, TX, USA) were used. The erythrocytes were washed by centrifuging at 1000× *g* for 5 min at 4 °C. The supernatant was discarded and the SRBC were diluted with GVB++ buffer (with Ca^2+^ and Mg^2+^, pH 7.3) (Complement Technology, Inc., Tyler, TX, USA). This buffer contains 0.1% gelatin, 5 mM Veronal, 145 mM NaCl, 0.025% NaN_3_, 0.15 mM CaCl_2_ and 0.5 mM MgCl_2_. This buffer composition is important for the classical pathway process. First, the control tubes were prepared (double determination): 0% lysis with GVBE buffer (90 μL) and 50% and 100% lysis with ddH_2_O (85 μL and 90 μL, respectively), as well as a positive control with 80 μL GVB++ and 10 μL of any serum. Subsequently, serial dilutions of the test serum were made (1:10, 1:20, 1:40, 1:80, 1:160, 1:320, 1:640). SRBC (10 μL) were added to each tube except for the 50% lysis control tube (5 μL). The tubes were incubated for 30 min at 37 °C. The hemolysis reaction was stopped by adding 200 μL ice-cold GVBE buffer (0.1% gelatin, 5 mM Veronal, 145 mM NaCl, 0.025% NaN_3_ and 10 mM EDTA), which inhibits the complement activation cascade, and centrifuged (1000× *g*, 3 min, 4 °C). The degree of hemolysis was quantified by measuring the absorbance of the hemoglobin released into the supernatant at 415 nm.

### 4.6. AH-50 Assay

The AH-50 is an assay to test the functional capability of the alternative complement components to lyse rabbit red blood cells (RRBC). Rabbit erythrocytes (Complement Technology, Inc.) were used for this assay. The erythrocytes were washed by centrifuging at 1000× *g* for 5 min at 4 °C. The supernatant was discarded and the RRBC were diluted with GVB^0^ (without Ca^2+^ or Mg^2+^, pH 7.3). This basic buffer for complement assays contains 0.1% gelatin, 5 mM Veronal, 145 mM NaCl and 0.025% NaN_3_. For the dilution series, GVB^0^ (47.5 mL) was mixed with 0.1M magnesium-ethylene glycol-bis(β-aminoethyl ether)-*N,N,N′,N′*-tetraacetic acid (MgEGTA) (2.5 mL). As Mg^2+^ is required for the activation of the alternative complement pathway, addition of MgEGTA allowed this activation. First, the control tubes were prepared (double determination): 0% lysis with GVBE buffer (27.5 μL) and serum (10 μL) and 50% and 100% lysis with ddH_2_O (43.5 μL and 37.5 μL, respectively), as well as a positive control with 27.5 μL GVB^0^+MgEGTA and 10 μL of any serum. Subsequently, serial dilutions of the test serum were made (1:2, 1:4, 1:8, 1:16, 1:32). Erythrocytes (12.5 μL) were added to each tube except for the 50% lysis control tube (6.25 μL). The tubes were incubated for 30 min at 37 °C. The hemolysis reaction was stopped by adding 200 µl ice-cold GVBE buffer (with EDTA, pH 7.3) (Complement Technology Inc.). GVBE buffer (again 0.1% gelatin, 5 mM Veronal, 145 mM NaCl, 0.025% NaN_3_, 10 mM EDTA) inhibited the complement activation and was subsequently centrifuged (1000× *g*, 3 min, 4 °C). The degree of hemolysis was quantified by measuring the absorbance of the hemoglobin released into the supernatant at 415 nm.

### 4.7. Statistical Analysis

Values are presented as mean ± standard error of the mean. Datasets were analyzed for normal distribution by the Shapiro–Wilk test. Significant differences between the groups were detected by the analysis of variance with Fisher LSD post hoc test, with a value of *p* ≤ 0.05 considered statistically significant.

## 5. Conclusions

Although dental disorders are a widespread problem in several species, there remains limited knowledge about the cells and cytokines involved, as well as the underlying pathological molecular mechanisms. Therefore, the present study was performed to reveal critically involved immunologic components in periodontitis and TR, with a specific focus on MCs and the complement system. We found that MCs, C3-positive cells and C5a-positive cells were increased in TR compared to control cats and cats suffering from gingivitis, indicating that all these are involved in the destruction of the dental mineralized matrix. MCs in the gingiva were demonstrated to express C3 and C5a. Complement system and MCs together with calcium crystals might lead to an activation of osteoclasts, resulting in the resorption of the tooth’s mineralized tissues and alveolar bone. Using the cat as a model of naturally occurring TR, this study may help to reveal new therapeutic strategies to prevent and treat this dental disorder, which not only occurs in animals, but also in humans.

## Figures and Tables

**Figure 1 ijms-22-02759-f001:**
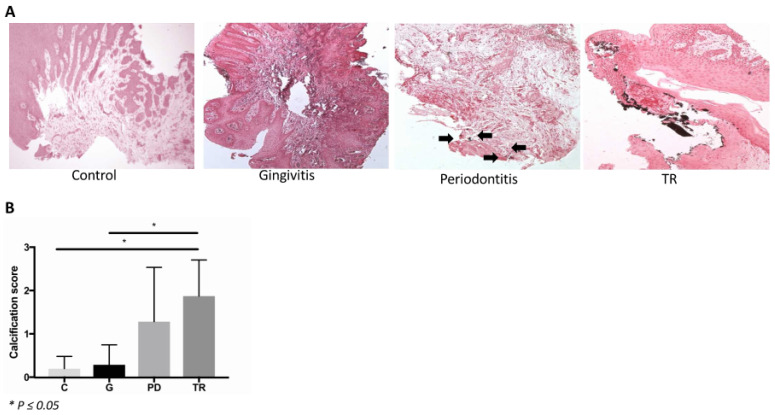
(**A**) Von Kossa staining of the gingiva in control cats (C) (*n* = 4) and cats affected by gingivitis (G) (*n* = 6), periodontitis (PD) (*n* = 8) or tooth resorption (TR) (*n* = 11). Images were obtained at 100× magnification. In control and gingivitis sections, no crystals were observed (calcification score 0). In periodontitis samples, rare, tiny crystals, marked by black arrows, were present (calcification score 1). In this TR section many large crystals could be seen (calcification score 3). (**B**) There was a significant increase in the occurrence of crystals in the tooth resorption (TR) (*n* = 11) group compared with the control (C) (*n* = 4) and gingivitis (G) (*n* = 6) groups (both *p* ≤ 0.05). There was no significant difference between the periodontitis group (PD) (*n* = 8) and the other groups.

**Figure 2 ijms-22-02759-f002:**
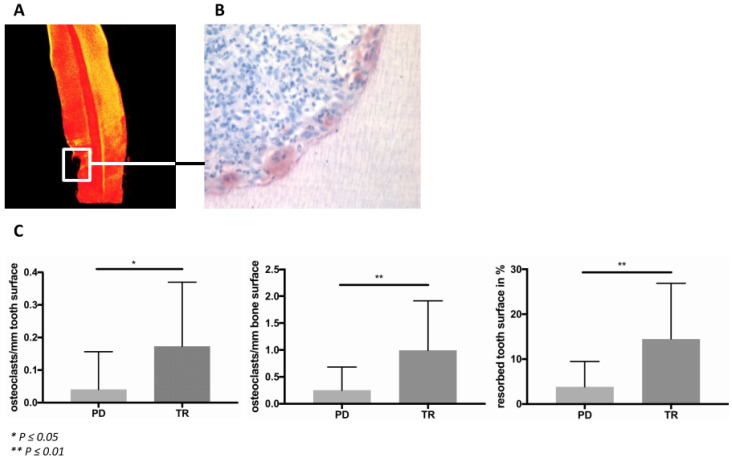
(**A**) Micro-Computed Tomography (μCT) image of a canine affected by tooth resorption (TR). The affected area is marked with a white rectangle. (**B**) Tartrate-resistant acid phosphatase staining of this tooth revealed many osteoclasts in this area, directly adjacent to the dentine of the tooth. The image was obtained at 400× magnification. (**C**) The evaluation showed significantly more osteoclasts in TR (*n* = 11) compared to periodontitis (PD) (*n* = 23) in both of the mineralized tissues, bone (*p* ≤ 0.01) and tooth (*p* ≤ 0.05). There was also a significant difference in the percentage of the eroded tooth surface between these groups (*p* ≤ 0.01).

**Figure 3 ijms-22-02759-f003:**
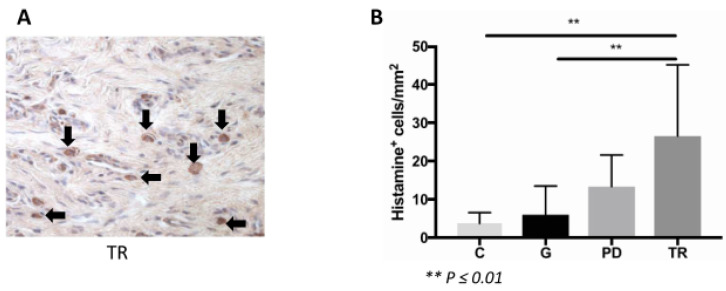
(**A**) Image of the immunohistochemical histamine staining in a TR section. The image was taken at 200× magnification. MCs could be identified. (**B**) Immunohistochemical staining for histamine showed a significant increase of MCs in the TR group (*n* = 11) compared with the control (*n* = 7) and gingivitis groups (*n* = 10) (both *p* ≤ 0.01). There was no difference between the TR and periodontitis groups (*n* = 8).

**Figure 4 ijms-22-02759-f004:**
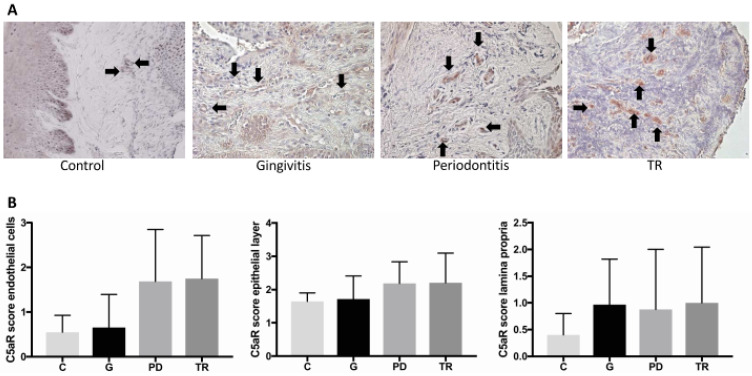
(**A**) Immunohistochemical staining of the complement C5a receptor (C5aR). Images were taken at 400× magnification. The endothelial layer of the vessels in the gingiva of the controls (C) (*n* = 4) and cats affected by gingivitis (G) (*n* = 6), periodontitis (PD) (*n* = 7) or tooth resorption (TR) (*n* = 12) is marked by black arrows. In the controls and gingivitis animals, only a few endothelial layers were stained. The staining in general was less marked than in in the other two groups (PD, TR). (**B**) No significant differences were observed between the groups. However, there was a trend for a difference in periodontitis and TR compared to control and gingivitis regarding the endothelial layer. In addition, there was a trend for a difference in positively stained cells of the lamina propria between the control group and the other groups.

**Figure 5 ijms-22-02759-f005:**
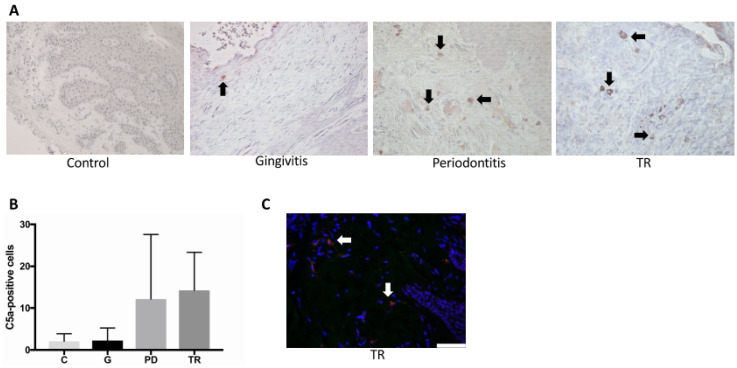
(**A**) Immunohistochemical staining of C5a in the gingiva of the controls (C) (*n* = 4) and of cats affected by gingivitis (G) (*n* = 6), periodontitis (PD) (*n* = 9) or tooth resorption (TR) (*n* = 12). Images were obtained at 400× magnification. No positive cells were observed in the control section. In the gingivitis section, a single positively stained cell was seen, marked by a black arrow. In the other two groups, notably more cells could be seen. (**B**) There was a trend for an increase in positively stained cells in the PD and TR groups compared with the other two groups. (**C**) Double immunofluorescence staining in the gingiva of a tooth affected by TR. Scale bar 50 μm. Mast cell granules (Avidin) were stained red, while C5a was stained green. Mast cells positive for C5a were observed, marked by white arrows.

**Figure 6 ijms-22-02759-f006:**
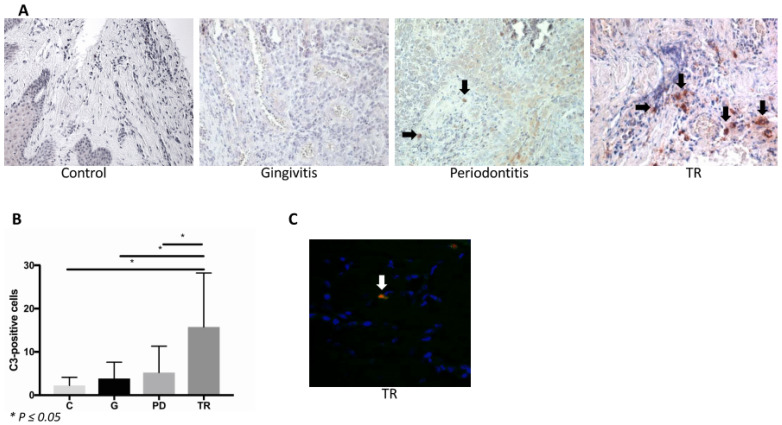
(**A**) Immunohistochemical staining of complement factor C3 in the gingiva of the controls (C) (*n* = 4) and cats affected by gingivitis (G) (*n* = 6), periodontitis (PD) (*n* = 9) or tooth resorption (TR) (*n* = 12). Images were obtained at 400× magnification. In the control and gingivitis groups, no positively stained cells were observed. Two C3-positive cells were seen in periodontitis, marked by black arrows. The greatest number of positively stained cells could be observed in TR. (**B**) There was a statistically significant increase in C3-positive cells in the TR group compared with the other three groups (*p* ≤ 0.05). (**C**) Double immunofluorescence staining in the gingiva of a tooth affected by TR. Scale bar 50 μm. Mast cell granules (Avidin) were stained red, while C5 was stained green. In this section, one mast cell was positive for C3, marked by a white arrow.

**Figure 7 ijms-22-02759-f007:**
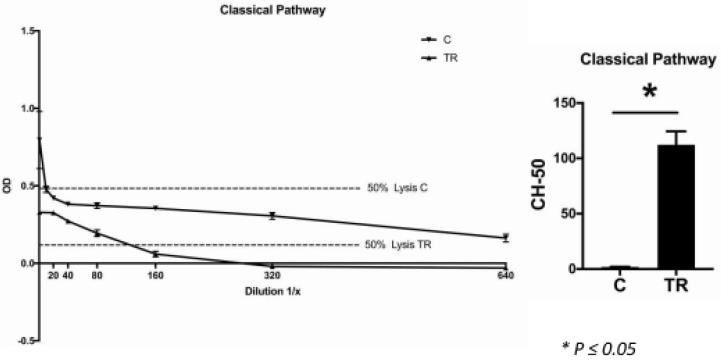
Hemolytic assay for the detection of complement classical pathway function. The optical density of the serum samples of both the control (C) and tooth resorption (TR) groups were plotted against the dilution factor. The curves show the control (4 cats) and TR groups (12 cats) with different levels of complement function. In TR, 50% lysis occurred between serum dilutions 1:80 and 1:160, in the control group between the undiluted serum and dilution 1:10. There was a significant difference in the serum complement concentration between the two groups, being significantly higher in the TR group than in the control group (*p* ≤ 0.05).

**Figure 8 ijms-22-02759-f008:**
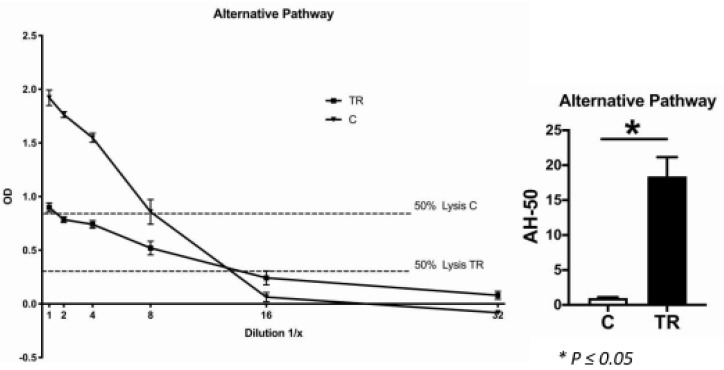
Hemolytic assay for the detection of complement alternative pathway function. The optical density of the serum samples of both the control (C) and tooth resorption (TR) groups were plotted against the dilution factor. The curves show the control (4 cats) and TR groups (10 cats) with different levels of complement function. In both groups, 50% lysis occurred between serum dilutions 1:8 and 1:16. Nevertheless, there was a significant difference in the serum complement concentration between the two groups, being significantly higher in the TR group than in the control group (*p* ≤ 0.05).

**Table 1 ijms-22-02759-t001:** Descriptive statistics of the anamnestic data. The cats were affected by gingivitis (G), periodontitis (PD) or tooth resorption (TR). The age of the cats, the sex, indoor only and dry food only were considered. Regarding the sex distribution, there was a statistically significant difference between the gingivitis and periodontitis groups (*p* ≤ 0.05).

	*n*	Age (Years)							Sex		Indoor Only	Dry Food Only
		<1	1–3	3–5	5–10	10–15	>15	Unknown	Male	Female		
**G**	5	2	2	-	-	1	-	-	5	0 *	3	1
**PD**	7	1	1	1	2	2	-	-	2	5 *	4	0
**TR**	16	-	2	-	4	1	1	8	7	9	10	1

* *p* ≤ 0.05.

**Table 2 ijms-22-02759-t002:** Overview of the included cats. These cats had an appointment for routine dental treatment and were included in the study. An overview of the obtained samples, age, sex, indoor only and dry food only is given.

Cat (*n* = 33)	Samples			Anamnestic Data										
	Gingiva	Teeth	Blood	Age (Years)							Sex		Indoor Only	Dry Food Only
				<1	1–3	3–5	5–10	10–15	>15	Unknown	Male	Female		
12	✓	-	-			✓						✓	✓	
14	✓	✓	-							✓		✓		
15	✓	-	-							✓		✓	✓	
16	✓	✓	-							✓		✓	✓	
19	✓	✓	-	✓								✓		
20	✓	✓	-		✓						✓			
22	✓	✓	-					✓				✓	✓	
23	✓	-	-					✓			✓			
24	✓	✓	-				✓					✓	✓	
27	-	✓	-					✓			✓			
28	✓	✓	-				✓					✓		
30	✓	✓	✓							✓		✓		
31	-	-	✓							✓		✓	✓	
33	-	✓	-					✓				✓	✓	
34	✓	✓	✓							✓	✓			
36	✓	✓	✓							✓	✓		✓	
37	-	-	✓			✓						✓	✓	
38	-	-	✓				✓				✓		✓	
39	✓	-	-	✓							✓		✓	
40	-	-	✓					✓			✓			✓
41	✓	-	✓				✓					✓	✓	
42	-	-	✓		✓							✓	✓	✓
43	✓	✓	-					✓				✓	✓	
44	✓	-	✓			✓						✓		
45	✓	✓	✓			✓					✓		✓	✓
47	✓	-	-		✓						✓		✓	✓
48	-	-	✓			✓						✓	✓	
49	✓	-	-	✓							✓		✓	
50	✓	-	-	✓							✓		✓	
52	✓	-	-							✓	✓		✓	
53	✓	-	-					✓			✓		✓	
54	✓	-	-		✓						✓			
55	✓	-	-		✓							✓		

## Data Availability

All data are included into this script.

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
