# Peer review of "Increased Presence of Complement Factors and Mast Cells in Alveolar Bone and Tooth Resorption"

_ijms, 2021, doi:10.3390/ijms22052759_

Round 1

Reviewer 1 Report

The authors have responded adequately to my raised concerns

Author Response

We thank the reviewer for the positive comment. 

Reviewer 2 Report

This is a very comprehensive large animal studies assessing the presence of mast cells and complement factor activities in gingival, tooth, and blood samples collected from cats with different conditions (gingivitis, periodontitis, and root resorption). Although the simple size has its limitation in its power, the trends in immunohistostaining is evident and the complement activities in the serum is striking. However, some comments for minor revision as follows: 1. How was the sample collected in each animal? Especially for those with root resorption. And how is the area of interest defined? Please laborate. 2. Why there is no hemolytic analysis for gingivitis or periodontitis group? 3. The title and conclusion can be further refine to reflect the results more precisely.

Author Response

This is a very comprehensive large animal studies assessing the presence of mast cells and complement factor activities in gingival, tooth, and blood samples collected from cats with different conditions (gingivitis, periodontitis, and root resorption). Although the simple size has its limitation in its power, the trends in immunohistostaining is evident and the complement activities in the serum is striking. However, some comments for minor revision as follows: 1. How was the sample collected in each animal? Especially for those with root resorption. And how is the area of interest defined? Please laborate.

Answer: We thank the reviewer for the positive comments and added further information about sample collection to the manuscript.

  1. Why there is no hemolytic analysis for gingivitis or periodontitis group?

Answer: Unfortunately, we were not able to withdraw blood from enough cats which suffer only from gingivitis and periodontitis to analyze CH-50 and AH-50. Therefore, we can not perform this recommended analysis with an appropriate sample size. We did an extensive literature research to find historical values for CH-50 or AH-50 in different cat populations. However, since the cat is a quite uncommon model for biomedical research, we were not able to find any data. However, the strength of cats as a model in our case is clearly, that the investigated disease occurs naturally in a high prevalence in cats and allows us to evaluate the presence of complement factors and mast cells in this clinically relevant setting. The next step would be to evaluate the potential benefit of mast cell or complement inhibitors in feline TR. Another benefit of using cats to study TR is that clinical trials can be much easier done than in humans, but results from these trails might be a great relevance for the human situation also.

  1. The title and conclusion can be further refine to reflect the results more precisely.

Answer: We further revised the title and the conclusion in the abstract.

Reviewer 3 Report

The authors are reporting the involvement of complement factors and mast cells in periodontal diseases. This report is showing interesting findings, but there are some points which should be revised.

1) This referee is thinking that the title should be changed to “The presence of complement factors-positive mast cells in feline alveolar bone and tooth resorption”. This is because the involvement of mast cells in alveolar bone and tooth resorption was already reported (Ref. 1).

2) In Figure 3, the authors utilized Toluidine blue staining for mast cell detection. Though, the stained cells are difficult to be identified as mast cells in Figure 3A included in this PDF file. This referee is thinking that the data using Toluidine blue staining can be deleted.

3) Currently, avidin staining for mast cell detection is thought to be unreliable. Double immunofluorescence staining using anti-Histamine and anti-C3 or C5 should be shown in Figures 5C and 6C.

END

Author Response

The authors are reporting the involvement of complement factors and mast cells in periodontal diseases. This report is showing interesting findings, but there are some points which should be revised.

1) This referee is thinking that the title should be changed to “The presence of complement factors-positive mast cells in feline alveolar bone and tooth resorption”. This is because the involvement of mast cells in alveolar bone and tooth resorption was already reported (Ref. 1).

Answer: The reviewer is right that the involvement of mast cells in alveolar bone and tooth resorption has been shown already, however this is still an important part of our manuscript to show that mast cells in the gingiva express complement receptors and might be involved in the disease progression. Therefore, we think this should be mentioned in the title.

2) In Figure 3, the authors utilized Toluidine blue staining for mast cell detection. Though, the stained cells are difficult to be identified as mast cells in Figure 3A included in this PDF file. This referee is thinking that the data using Toluidine blue staining can be deleted.

Answer: The reviewer is right that Toluidine blue staining does not add more information. Therefore we deleted these data.

3) Currently, avidin staining for mast cell detection is thought to be unreliable. Double immunofluorescence staining using anti-Histamine and anti-C3 or C5 should be shown in Figures 5C and 6C.

Answer: The reviewer is right that Avidin staining is not specific for mast cell surface structures, but rather specific for mast cell granules. To prove this specificity, we used sections from wildtype mast cell competent and Mcpt5-Cre+ R-DTAflox/flox mast cell deficient mice to establish Avidin staining. Bone marrow sections from Mast cell deficient mice did not show any staining with Avidin. We further added these information to the manuscript and revised the text that Avidin stains mast cell granules rather than mast cell surface structures.

This manuscript is a resubmission of an earlier submission. The following is a list of the peer review reports and author responses from that submission.

Round 1

Reviewer 1 Report

In this study, Widmann et al have studied the presence of osteoclasts, complement factors, mast cells and calcium crystals in cats suffering from tooth resorption, gingivitis or periodontitis. An advantage of using the feline species for this purpose is that cats have a high prevalence of such disorders. The authors show that, in particular, tooth resorption is accompanied by increases in the mast cell population, increased presence of calcium crystals, increased numbers of osteoclasts, as well as increased levels of C5a and C3. The observations are of some interest, but the study is overall very descriptive. Moreover, the various findings are quite disconnected. For example, the authors do not provide any link between the increases in the mast cell populations and the increased positivity for C5a/C3.

Additional points

-Fig. 2. Here it is shown that there is an increased presence of osteoclasts in tooth resorption vs. periodontitis. However, the data for control and gingivitis are lacking, and should be added.

-Fig. 3. Panel B shows staining for histamine in tooth resorption. Please also show staining and quantification of data for control, gingivitis and periodontitis.

-Fig. 5.  Here it is shown that there is a trend of increased positivity for C5a in tooth resorption and periodontitis vs. control and gingivitis. It would be valuable if the C5a-positive cells could be identified.

-Fig. 6. This figure reveals a significant increase in positivity for C3 in tooth resorption vs. control, gingivitis and periodontitis. However, the cell type(s) positive for C3 are not identified- this would be valuable information. Further, if would be more informative if it could be shown whether this positivity is due to the presence of noncleaved C3, C3a or C3b.

-Fig. 7, 8. Data for gingivitis and periodontitis should be shown.

-The linguistic quality of the manuscript can be improved.